# Development of a Laboratory Risk-Score Model to Predict One-Year Mortality in Acute Myocardial Infarction Survivors

**DOI:** 10.3390/jcm11123497

**Published:** 2022-06-17

**Authors:** Yuhei Goriki, Atsushi Tanaka, Goro Yoshioka, Kensaku Nishihira, Nehiro Kuriyama, Yoshisato Shibata, Koichi Node

**Affiliations:** 1Department of Cardiovascular Medicine, National Hospital Organization Ureshino Medical Center, Ureshino 843-0393, Japan; i03.eou2@gmail.com; 2Department of Cardiovascular Medicine, Saga University, Saga 849-8501, Japan; s04211090s@gmail.com (G.Y.); node@cc.saga-u.ac.jp (K.N.); 3Miyazaki Medical Association Hospital Cardiovascular Center, Miyazaki 880-0834, Japan; nishihira@med.miyazaki-u.ac.jp (K.N.); n-kuriyama@cure.or.jp (N.K.); yshibata@cure.or.jp (Y.S.)

**Keywords:** biomarker, mortality, myocardial infarction, risk-score model

## Abstract

The high post-discharge mortality rate of acute myocardial infarction (AMI) survivors is concerning, indicating a need for reliable, easy-to-use risk prediction tools. We aimed to examine if a combined pre-procedural blood testing risk model predicts one-year mortality in AMI survivors. Overall, 1355 consecutive AMI patients who received primary coronary revascularization were divided into derivation (*n* = 949) and validation (*n* = 406) cohorts. A risk-score model of parameters from pre-procedural routine blood testing on admission was generated. In the derivation cohort, multivariable analysis demonstrated that hemoglobin < 11 g/dL (odds ratio (OR) 4.01), estimated glomerular filtration rate < 30 mL/min/1.73 m^2^ (OR 3.75), albumin < 3.8 mg/dL (OR 3.37), and high-sensitivity troponin I > 2560 ng/L (OR 3.78) were significantly associated with one-year mortality after discharge. An increased risk score, assigned from 0 to 4 points according to the counts of selected variables, was significantly associated with higher one-year mortality in both cohorts (*p* < 0.001). Receiver-operating characteristics curve analyses of risk models demonstrated adequate discrimination between patients with and without one-year death (area under the curve (95% confidence interval) 0.850 (0.756–0.912) in the derivation cohort; 0.820 (0.664–0.913) in the validation cohort). Our laboratory risk-score model can be useful for predicting one-year mortality in AMI survivors.

## 1. Introduction

Acute myocardial infarction (AMI) is a major cause of poor outcomes and clinical concerns worldwide [1]. Over the past two decades, in-hospital death rates from AMI have decreased dramatically, partly due to advances in the clinical management of the acute phase of AMI and guideline-directed medical therapy [2]. However, long-term prognosis of AMI survivors is still unfavorable. In this regard, post-discharge mortality remains a clinical concern [2,3]. Therefore, there is a need for reliable and easy-to-use risk prediction tools for early identification of at-risk patients, which may help with timely prevention and well-tailored treatment.

Several risk-score models to predict prognosis after acute coronary syndrome (ACS), including AMI, are currently available [4,5]. The Global Registry of Acute Coronary Events (GRACE) 2.0 score is one of the most established risk-score models for determining mortality risk in AMI patients [4]. This model was created prior to the contemporary era of optimal medical therapy and increased usage of percutaneous coronary intervention (PCI) for AMI patients [2,6]. In addition, this model requires several clinical variables: age, systolic blood pressure (BP), heart rate, creatinine, cardiac arrest at admission, ST-segment deviation, abnormal cardiac enzyme, and Killip classification. However, in the emergency setting of AMI, several variables, such as heart rate and systolic BP, often fluctuate significantly, which could compromise the prediction value of the GRACE 2.0 model, resulting in a requirement for reassessment. Unlike these clinical measurements, blood parameters can be rapidly obtained in a non-subjective fashion, even in the emergent setting of AMI, suggesting that a blood-parameter-based model may be easy to use for AMI mortality risk prediction.

Several models of combined blood variables have been used as predictive indicators of AMI mortality risk [7,8]. We have also previously reported a risk-score model combining pre-procedural laboratory variables to predict the risk of in-hospital death in ST-segment elevation myocardial infarction (STEMI), comparable to the GRACE 2.0 model [9]. However, whether combined-blood-parameter-based models could predict post-discharge mortality in AMI survivors remains largely unknown. Herein, we aimed to create a risk-score model based on a combination of pre-procedural laboratory parameters for one-year mortality after discharge in AMI survivors and compare its predictive ability with a conventional model (GRACE 2.0 model).

## 2. Materials and Methods

### 2.1. Design and Participants

This was a retrospective observational study conducted in Miyazaki Medical Association Hospital. The study population comprised 1852 consecutive patients hospitalized for ACS between Apr 2012 and Jan 2018, were included in the present study. Patients who did not undergo primary PCI, recurrent ACS or unstable angina pectoris, who died during hospitalization, lost to follow-up one year after discharge, and lack of laboratory information on admission were excluded from the analyses. Thus, 1355 patients were included in the present study. Patients were randomly classified into either derivation (*n* = 949) or validation cohorts (*n* = 406) [10,11] (Figure 1). This study was approved by the Institutional Review Board of Miyazaki Medical Association Hospital and complied with the latest Helsinki Declaration. Nevertheless, written informed consent was waived due to the retrospective of the study.

### 2.2. Definition of STEMI and NSTEMI

STEMI and non-ST-segment elevation myocardial infarction (NSTEMI) were diagnosed by cardiologists based on the universal definitions [12]. Treatment and management depend on the latest domestic guidelines released by the Japanese Circulation Society (Diagnosis and Treatment of Acute Coronary Syndrome).

### 2.3. Data Acquisition

Participants’ baseline characteristics and clinical manifestations, vital signs, medical history, usual laboratory data, high-sensitivity troponin I (hsTnI), type of AMI, Killip classification and left ventricular ejection fraction were collected on admission. Hs-TnI levels were measured on a chemiluminescence immunoassay (ARCHITECT^®^ high sensitive troponin I (Abbott Japan, Tokyo, Japan)) with a coefficient of variation < 10% at 32 ng/L and 99th percentile reference limit < 34.2 or 15.6 ng/L (male or female). For the procedure-related parameters, clinical information on peak creatine kinase (CK), culprit lesion and mechanical support were collected during coronary procedure. Medications at discharge were also collected. Information on post-discharge death was collected by medical records or telephone calls. The estimated glomerular filtration rate (eGFR) was calculated with the revised equation for the Japanese population [13].

### 2.4. Statistics

Continuous variables are expressed as mean ± standard deviation for normal distribution or median [interquartile range] for non-normal distribution values. Categorical variables are shown as numbers (%). Comparisons of continuous variables between both cohorts were done with Student’s *t*-test or Wilcoxon test, where appropriate. Categorical variables were compared using the chi-squared test.

The laboratory variables significantly associated with post-discharge death selected by the univariate analysis were categorized based on the cutoff values reported previously [14,15,16,17,18,19,20,21] and then applied for the multivariable analysis to develop the risk-score model. Those variables were further selected using a multivariable logistic regression model using the backward factor elimination method. Finally, the remaining variables were given an equivalent point to calculate the risk score for one-year mortality. The subjects were classified into three groups based on the total scores, as follows: low risk (0–1 point), moderate risk (2 points) and high risk (3–4 points). Cochran-Armitage trend analysis was used to assess statistical trends among three risk groups. The predictive abilities of the risk models for predicting post-discharged death were assessed for their discrimination and calibration, and which were analyzed by the receiver operating characteristic curve and Hosmer–Lemeshow goodness-of-fit test, respectively. The risk score for predicting one-year mortality was also calculated using the GRACE 2.0 ACS Risk Calculator app. We estimated the area under the curve (AUC) of the GRACE 2.0 model and compared it with that of the present model derived from the validation cohort. Differences in those AUCs were appraised by the DeLong method [22]. Statistical analyses were conducted using the JMP version 15 (SAS Institute Inc., Cary, NC, USA), and statistical significance was set at *p*-value < 0.05 (2-tailed), except for the Hosmer–Lemeshow goodness-of-fit test.

## 3. Results

### 3.1. Baseline Demographics and Characteristics

The study population consisted of 1355 subjects (derivation 949; validation 406). Baseline demographics and characteristics of the two cohorts are listed in Table 1. There were no significant differences in the clinical information on admission and during hospitalization between the derivation and validation cohorts. Medications at discharge were comparable between the two cohorts. Post-discharge deaths were observed in 30 patients (3.2%) in the derivation cohort and 14 (3.5%) in the validation cohort.

### 3.2. Laboratory Parameters Associated with Post-Discharge Death

Table 2 shows the univariate analysis of blood variables between survivors and non-survivors at one-year post-discharge in the derivation cohort. Significant variables detected in the univariate analysis were subjected to a multivariable stepwise backward logistic regression analysis. In that analysis, hemoglobin level < 11 g/dL, eGFR < 30 mL/min/1.73 m^2^, albumin level < 3.8 mg/dL, and hs-TnI > 2560 ng/L (normal upper limit × 80) were significantly associated with post-discharge death in the derivation cohort (Table 3). The odds ratio for one-year mortality ranged from 3.37 to 4.01. Zero to four points were assigned to each patient according to the number of risk factors they had.

### 3.3. Predictive Model of Post-Discharge Death

The incidence of post-discharge death during one-year follow-up increased significantly as the total risk score elevated in both cohorts (Figure 2A,B). The risk-score model demonstrated adequate discrimination between subjects who died or not after discharge in the validation (AUC, 95% confidence interval (CI): 0.850, 0.756–0.913) and derivation (0.820, 0.664–0.913) cohorts (Figure 3). The Hosmer–Lemeshow test indicated a favorable fit in both cohorts (χ^2^ = 0.328, *p* = 0.849 for the derivation; χ^2^ = 0.556, *p* = 0.757 for the validation). When patients were further classified into three subgroups based on their risk score: 0–1 point (defined as low-risk), 2 points (moderate-risk), and 3–4 points (high-risk), a similar trend for post-discharge mortality during one-year follow-up was also observed in those subgroups (Figure 4A,B).

### 3.4. Comparison with GRACE 2.0 Model

The AUCs of the present and GRACE 2.0 models in the validation cohort were 0.820 (95% CI, 0.664–0.913) and 0.806 (95% CI, 0.681–0.890). The predictive power was similar between the two models (Figure 5). Additionally, we compared the predictive ability between these risk models based on the type of AMI and gender. In all cases, the predictive power was not significantly different between these models (Table 4). Furthermore, the laboratory model was able to stratify the possible risk of post-discharge death, especially in the high-risk subgroup from the GRACE 2.0 model (risk-score > 8.0%) [23], but not in the low–intermediate-risk groups from the GRACE 2.0 model (risk-score ≤ 8.0%) (Figure 6).

## 4. Discussion

The main findings of this investigation were as follows: (i) individual blood variables measured on admission (hemoglobin < 11 g/dL, eGFR < 30 mL/min/1.73 m^2^, albumin < 3.8 mg/dL, and hs-TnI > 2560 ng/L) were independently related with an augmented risk of post-discharge mortality rates at one-year; (ii) a simple model, using a combination of pre-procedural laboratory measures, can be useful for assessing the risk of post-discharge death at one-year follow-up; (iii) the predictive ability of our model was similar to that of the GRACE 2.0 model; and (iv) our model provided predictive power to further subdivide the high-risk population estimated by the GRACE 2.0 model. Therefore, these results may indicate that our novel model with pre-procedural laboratory parameters can help predict one-year mortality in AMI survivors.

Some risk stratification models for estimating the risk of post-discharge death rates have been developed for patients with AMI. Among these risk prediction models, the GRACE 2.0 system has been recommended for stratifying AMI mortality risk according to guidelines [24]. However, cohorts enrolled from the GRACE model were patients in the 2000s, while medical management of AMI has developed beyond the clinical surroundings of the 2000s [2]. Moreover, some of the hemodynamic statuses required to calculate the GRACE score often fluctuate widely, especially in the emergency clinical phase of AMI. Therefore, the risk estimated by the GRACE model may also vary. While the measurements of blood parameters can be performed readily and objectively, this study sought to create a laboratory-based model to estimate the risk of one-year death in AMI survivors.

Individual blood parameters considered for the risk assessment in the present study are useful markers for predicting the prognosis in patients with AMI [14,15,16,17,18,19,20,21]. In particular, the presence of anemia or renal dysfunction is a powerful predictor of poor outcomes in post-AMI patients [7,8,9,10]. Actually, several risk calculators for predicting long-term mortality need renal functional parameters and hemoglobin levels [4,5]. Besides anemia- and chronic kidney disease (CKD)-related parameters, our model found two new individual blood parameters as possible candidates to predict post-discharge death: albumin and hsTnI levels.

Albumin is a marker of nutrition, frailty, and inflammation [25], all of which have been individually reported to contribute to the cardiovascular disease prognosis [26,27]. Recently, the relationship between albumin level and post-discharge prognosis of AMI has been reported [28,29]. Thus, serum albumin levels are affected by various aspects of clinical situations and may represent a predictive marker for clinical outcomes in post-AMI patients.

Our study also demonstrated a relationship between hsTnI and one-year death rates in AMI survivors. The biological kinetics of troponin on admission due to AMI was associated with ischemic time, infarct size and death during hospitalization [21,30]. Additionally, left ventricular dysfunction and onset-to-balloon time were predictors of cardiovascular events after AMI [31]. Therefore, the severity of myocardial damage, as measured by hsTnI on admission, has the potential to predict the incidence of post-discharge death in AMI survivors. In our study, the multivariable logistic regression analysis eventually selected those four parameters and co-included them in our model, which provided the predictive power of post-discharge death in AMI survivors, similarly to the GRACE 2.0 model.

In 2015, Pocock et al. created a predictive model for one-year mortality in AMI [5]. That model comprised 12 clinical parameters and discriminated the risk of post-discharge death within one year after AMI. However, calculating risk may be complicated, because as many as 12 factors are required, hampering the dissemination to clinical practice, especially in the emergency setting. Furthermore, the predictive ability of that model in comparison with other models was also unknown. In contrast, our model is easy to calculate only with four variables immediately obtained at admission for AMI. Additionally, our model was able to predict the risk of post-AMI death one year after discharge, being comparable to that of the GRACE 2.0 model. Notably, the present model was useful for stratifying risk in the high-risk subpopulation classified by the GRACE 2.0 model. These findings suggest that our model is clinically helpful in improving the predictive value for the risk of post-discharge death after AMI, specifically in the high-risk population derived from the GRACE 2.0 model, simply and objectively.

Compared to those existing models, our study’s strengths and novelty were that we developed the risk-score model showing the predictive ability comparable to the GRACE 2.0 model by combining only four blood parameters, each of which has prognostic evidence in patients with AMI. Considering that each blood parameter can reflect different aspects of a patient’s medical conditions, combining those parameters could provide a comprehensive and integrated approach to risk stratification and predicting prognosis. Several models based on the combined blood parameters have predicted short-term clinical outcomes in patients with AMI [7,8,32]. To the best of our knowledge, we first show that a risk-score model based on the combination of blood parameters on admission for AMI can predict the mid-term prognosis in AMI survivors.

Our study has some potential limitations. Firstly, this study was not a multi-center, prospective design. Secondly, since the cohorts included only Japanese and in relatively small numbers, the generalizability of our findings to other ethnicities remains uncertain. Thirdly, the current analyses assessed the predictive power of a risk-score model composed of only pre-procedural blood parameters upon admission for AMI. In addition, our predictive model did not account for the laboratory parameters obtained at post-procedure and/or discharge. Therefore, we cannot determine whether the selected parameters dominantly reflect acutely evoked pathophysiological reactions due to AMI or the chronic clinical conditions of the patients. Fourthly, the platelet count was lower in non-survivors than in the survivors in the univariate analysis. Actually, the risk of bleeding complications may be augmented in patients with a lower level of platelet count by receiving antiplatelet therapy, adversely affecting prognosis. Conversely, the patients who underwent PCI for coronary artery disease should receive antiplatelet therapy according to the relevant guidelines to reduce the risk of stent thrombosis [33]. Accordingly, most subjects received that therapy upon discharge, and their prognoses with and without it were not compared in this study. Therefore, the possibility that the antiplatelet therapy upon discharge had affected prognosis to some extent in this study cohort cannot be excluded. Finally, the study focused only on the laboratory variables to develop the current risk-score model. Therefore, non-laboratory variables, such as age, vital signs and cardiac function, related to the prognosis after AMI were not considered to predict the risk of post-discharge death. Nevertheless, our laboratory-based model showed comparable performance to the GRACE 2.0 model in predicting post-discharge death and partly improved the risk stratification, specifically in the high-risk population derived from the conventional model. As the present study sought to create a laboratory-based model to predict one-year mortality after AMI, further research is required to assess whether our model can predict longer-term prognosis and/or other clinical outcomes after AMI.

## 5. Conclusions

Our findings suggest that the present risk-score model is useful for predicting one-year mortality in AMI survivors who underwent primary PCI simply and objectively.

## Figures and Tables

**Figure 1 jcm-11-03497-f001:**
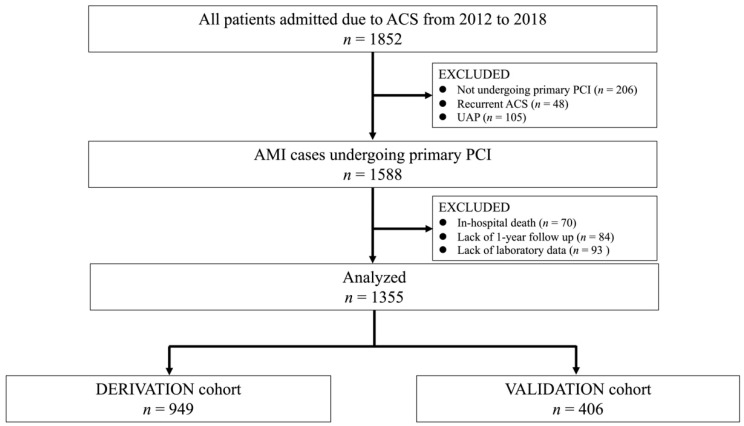
Flow chart of study participant selection. ACS, acute coronary syndrome; PCI, percutaneous coronary intervention; UAP, unstable angina pectoris.

**Figure 2 jcm-11-03497-f002:**
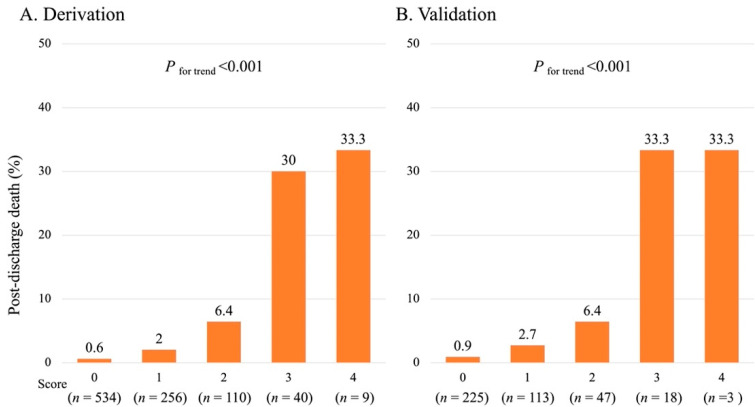
Post-discharge death rates according to the risk-score estimated by the laboratory model.

**Figure 3 jcm-11-03497-f003:**
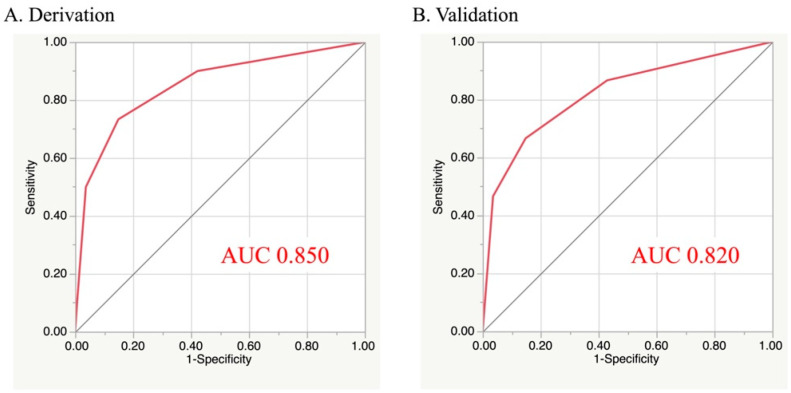
Receiver operating characteristic curves of the present model. AUC, area under the curve.

**Figure 4 jcm-11-03497-f004:**
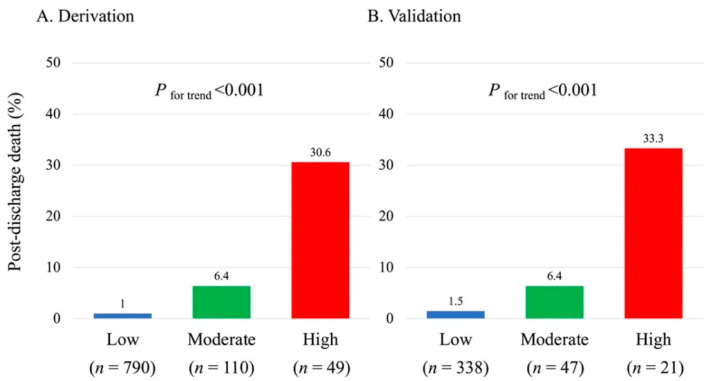
Incidence of post-discharge death in the risk-based subgroups. A total risk score of 0–1 point was defined as low-risk, 2 points as moderate-risk, and 3–4 points as high-risk.

**Figure 5 jcm-11-03497-f005:**
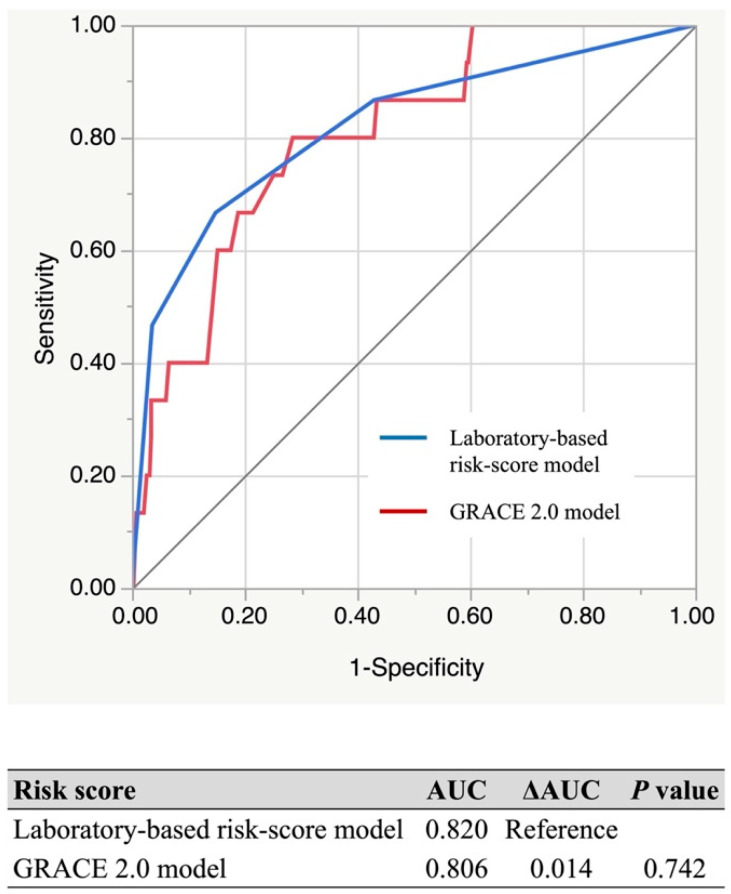
Comparison of the predictive abilities for one-year mortality between the laboratory and GRACE 2.0 models. AUCs of the laboratory (blue) and GRACE 2.0 (red) models in the validation cohort were 0.820 (95% CI, 0.664–0.913) and 0.810 (95% CI, 0.681–0.890). AUC, area under the curve; GRACE, Global Registry of Acute Coronary Events.

**Figure 6 jcm-11-03497-f006:**
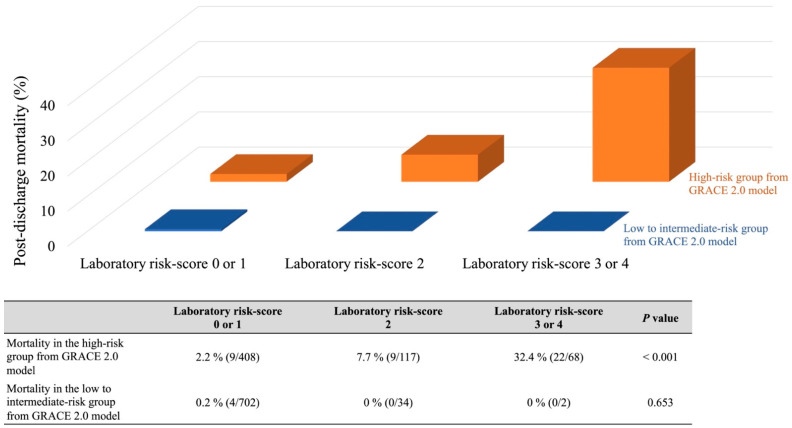
Dual-stratification by the laboratory model × the GRACE 2.0 model. All subjects from both study cohorts were stratified by the GRACE 2.0 model (high-risk > 8.0%, low to intermediate-risk ≤ 8.0%) and further subdivided into three subgroups by the laboratory model (low risk: 0–1 points, moderate risk: 2 points, and high risk: 3–4 points). The table below the graph shows the post-discharge one-year mortality for each subgroup. GRACE, The Global Registry of Acute Coronary Events.

**Table 1 jcm-11-03497-t001:** Demographics and characteristics in both cohorts.

	Derivation (*n* = 949)	Validation (*n* = 405)	*p*-Value
Age, years	69.2 ± 12.1	68.6 ± 12.6	0.435
Female, *n* (%)	250 (26.3)	111 (27.4)	0.635
Body mass index, kg/m^2^	24.0 ± 3.7	24.0 ± 4.1	0.789
Systolic blood pressure, mmHg	142.1 ± 28.5	141.0 ± 28.0	0.464
Pulse rate, bpm	77.1 ± 19.7	77.3 ± 17.8	0.863
Medical history			
Hypertension, *n* (%)	687 (72.4)	277 (68.4)	0.149
Dyslipidemia, *n* (%)	508 (53.5)	211 (52.1)	0.628
Diabetes mellitus, *n* (%)	264 (27.8)	102 (25.2)	0.317
Smoking, *n* (%)	467 (49.2)	189 (46.7)	0.391
Previous myocardial infarction, *n* (%)	70 (7.4)	23 (5.7)	0.274
Peripheral artery disease, *n* (%)	29 (3.1)	13 (3.2)	0.855
Malignancy, *n* (%)	45 (4.7)	14 (3.5)	0.288
Laboratory parameters			
White blood cell, ×10^2^/μL	96.4 ± 37.1	96.8 ± 33.3	0.857
Hemoglobin, g/dL	13.8 ± 2.1	13.8 ± 2.1	0.845
Platelet, ×10^4^/μL	21.7 ± 6.2	22.1 ± 6.3	0.246
Glycated hemoglobin A1c, %	6.0 (5.7, 6.6)	5.9 (5.6, 6.5)	0.168
Glucose, mg/dL	148 (107, 189)	141 (102, 181)	0.196
eGFR, mL/min/1.73 m^2^	66.2 ± 22.7	67.9 ± 24.1	0.899
LDL-cholesterol, mg/dL	122.8 ± 35.1	123.6 ± 37.7	0.698
HDL-cholesterol, mg/dL	46.8 ± 12.0	48.8 ± 13.2	0.097
Albumin, g/dL	4.0 ± 0.5	4.0 ± 0.5	0.648
Creatine kinase, U/L	156 (96, 356)	169 (100, 395)	0.286
hs-TnI, ng/L	300 (50, 3180)	380 (60, 3010)	0.653
STEMI, *n* (%)	640 (67.4)	278 (68.5)	0.709
Killip classification ≥ 3, *n* (%)	64 (6.7)	19 (4.7)	0.146
LVEF (on admission), %	52.4 ± 11.7	52.5 ± 10.6	0.550
Peak creatine kinase, IU/L	1231 (358, 2950)	1185 (299, 2687)	0.337
IABP, *n* (%)	93 (9.8)	28 (6.9)	0.086
ECMO, *n* (%)	9 (1.0)	2 (0.5)	0.166
Medication at discharge			
Antiplatelet therapy, *n* (%)	940 (99.0)	404 (99.5)	0.214
Aspirin (100 mg daily), *n* (%)	899 (94.7)	391 (96.5)	0.131
Prasugrel (3.75 mg daily), *n* (%)	208 (21.9)	73 (18.0)	0.221
Clopidogrel (75 mg daily), *n* (%)	670 (70.6)	308 (76.0)	0.089
Dual antiplatelet therapy, *n* (%)	857 (90.3)	377 (93.0)	0.165
Statin, *n* (%)	860 (90.6)	370 (91.1)	0.844
β-Blocker, *n* (%)	424 (44.9)	195 (48.0)	0.326
ACE inhibitor, *n* (%)	390 (41.1)	147 (36.2)	0.091
ARB, *n* (%)	296 (31.2)	147 (36.2)	0.074
Diuretic, *n* (%)	177 (18.7)	68 (16.8)	0.401
Post-discharge death during one-year follow-up, *n* (%)	30 (3.2)	14 (3.5)	0.785

Categorical variables are shown as numbers (%); data for continuous variables are shown as mean ± standard deviation for normal distribution or median (interquartile range) for non-normal distribution. ACE, angiotensin-converting enzyme; ARB, angiotensin receptor blocker; ECMO, extracorporeal membrane oxygenation; eGFR, estimated glomerular filtration rate; HDL, high-density lipoprotein; hs-TnI, high-sensitivity troponin I; IABP, intra-aortic balloon pumping; LDL, low-density lipoprotein; LVEF, left ventricular ejection fraction; STEMI, ST-segment elevation myocardial infarction.

**Table 2 jcm-11-03497-t002:** Univariate analysis of laboratory variables associated with post-discharge death.

	Survivors	Non-Survivors (Post-Discharge Death)	*p*-Value
White blood cell, ×10^3^/μL	9.6 ± 3.5	9.4 ± 4.2	0.656
Hemoglobin, g/dL	13.9 ± 2.0	11.7 ± 2.1	<0.001
Platelet, ×10^4^/μL	21.8 ± 6.1	18.1 ± 7.3	0.001
Glycated hemoglobin A1c, %	6.0 (5.4, 6.6)	6.3 (5.4, 6.7)	0.688
Glucose, mg/dL	148 (123, 187)	154 (119, 212)	0.531
eGFR, mL/min/1.73 m^2^	66.9 ± 22.1	43.7 ± 28.1	<0.001
LDL-cholesterol, mg/dL	123.7 ± 34.8	93.9 ± 32.9	<0.001
HDL-cholesterol, mg/dL	47.0 ± 12.1	41.8 ± 10.7	0.020
Albumin, mg/dL	4.1 ± 0.5	3.4 ± 0.5	<0.001
Creatine kinase, U/L	152 (96, 355)	232 (77, 855)	0.128
hs-TnI, ng/L	280 (50, 2030)	5900 (560, 21300)	<0.001

Data for continuous variables are expressed as mean ± standard deviation for normal distribution or median (interquartile range) for non-normal distribution. See Table 1 for abbreviation definitions.

**Table 3 jcm-11-03497-t003:** Independent predictors for post-discharge death and given risk-score.

	Odds Ratio	95% Confidence Interval	*p*-Value	Risk-Score
Hemoglobin < 11 g/dL	4.01	1.65–9.72	0.002	1
eGFR < 30 mL/min/1.73 m^2^	3.75	1.53–9.19	0.004	1
Albumin < 3.8 mg/dL	3.37	1.31–8.67	0.012	1
hs-TnI > 2560 ng/L (normal upper limit × 80)	3.78	1.64–8.72	0.002	1

See Table 1 for abbreviation definitions.

**Table 4 jcm-11-03497-t004:** Comparisons of AUCs between the two models according to AMI and sex statuses.

		AUC of the Laboratory-Based Risk-Score Model	AUC of the GRACE 2.0 Model	ΔAUC	*p*-Value
Type of AMI	STEMI	0.820	0.866	−0.046	0.124
	NSTEMI	0.871	0.855	0.016	0.738
Sex	Male	0.831	0.861	−0.036	0.397
	Female	0.836	0.840	−0.005	0.905

AMI, acute myocardial infarction; AUC, area under the curve; NSTEMI, non-ST-segment elevation myocardial infarction; STEMI, ST-segment elevation myocardial infarction. For other abbreviations, see Table 1.

## Data Availability

The raw data for the study will not be shared.

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
