# Peer review of "Development of a Laboratory Risk-Score Model to Predict One-Year Mortality in Acute Myocardial Infarction Survivors"

_jcm, 2022, doi:10.3390/jcm11123497_

Round 1
Reviewer 1 Report
The manuscript introduced a new model to predict one-year mortality in AMI survivors who underwent primary PCI. This new and laboratory parameters based model, with similar predictive ability to that of the GRACE 2.0 model, avoids some disadvantage of the GRACE 2.0 model, such as fluctuated hemodynamic statuses and outdated clinical surroundings of the 2000s. Although the results were obtained from small number and single center-Japanese, this new model is helpful in prediction of one-year mortality of AMI survivors and may be an important complement to the current GRACE 2.0 model.
Some minor concerns
Line 192 typing mistake “prdictive”should be“predictive”
Line 221 “cotribute” to “contribute”
Author Response
Response to the Reviewer 1’s Comments
Firstly, we are very grateful for the careful review and positive feedback on our manuscript. We have carefully revised the manuscript according to your valuable comments.
Some minor concerns
Line 192 typing mistake “prdictive”should be“predictive”
Line 221 “cotribute” to “contribute”
Response 1: Thank you so much for pointing out these mistakes. We have corrected these misspelled words.
Reviewer 2 Report
In the manuscript titled ‘Development of a Laboratory Risk-Score Model to Predict One-year Mortality in Acute Myocardial Infarction Survivors’ Goriki et al., have assessed the effectiveness of laboratory parameters on post-discharge mortality rate of acute myocardial infarction (AMI) survivors in a 1-year follow up, and have come up with a risk-score model of blood parameters from pre-procedural routine testing conducted upon admission in n=1355 AMI patients. The patients were subsequently separated and considered as part of derivation (n=949) and validation (n=406) cohorts. The investigators conclude that hemoglobin <11 g/dL, eGFR <30 mL/min/1.73 m2, albumin <3.8 mg/dL, and hsTnI >2.56 ng/mL were independently related with an increased risk of post-discharge mortality over a 1 year follow up. Comparing their approach with the GRACE 2.0 model they suggested that its predictive ability was similar to that of the GRACE 2.0 model.
The manuscript is well written, elaborate enough in the methods section clarifying the study design with a flow chart which is useful. The introduction and discussion sections place the investigation in specific clinical context. This being a retrospective study I have only a few suggestions to make which might add additional insights which have not been presented in the current manuscript.
1. Please elaborate the details of antiplatelet therapy (type and dosage) in the baseline characteristics table 1 since 99% of both derivation and validation cohorts were on anti-platelet therapy.
2. Platelet count had a significant impact on post discharge death in univariate analysis. Please consider the anti-platelet medications administered during PCI and upon discharge. How might anti-platelet therapy upon discharge have affected prognosis in this cohort? Due consideration should be given to this aspect.
3. Please add a separate paragraph highlighting the novelty of this study, since the parameters considered for risk assessment are quite well acknowledged.
4. In table 1 please replace ‘laboratory’ with ‘laboratory parameters’ and ‘antiplatelet’ with ‘antiplatelet therapy’.
Author Response
Response to the Reviewer 2’s Comments
Firstly, we are very grateful for the opportunity to revise our manuscript. We have carefully revised the manuscript according to your valuable comments and suggestions.
Comment 1. Please elaborate the details of antiplatelet therapy (type and dosage) in the baseline characteristics table 1 since 99% of both derivation and validation cohorts were on anti-platelet therapy.
Response 1: Thank you for the important and helpful suggestion. We have added the details of antiplatelet therapy to Table 1.
Comment 2. Platelet count had a significant impact on post discharge death in univariate analysis. Please consider the anti-platelet medications administered during PCI and upon discharge. How might anti-platelet therapy upon discharge have affected prognosis in this cohort? Due consideration should be given to this aspect.
Response 2: Thank you very much for this important comment. As pointed out, univariate analysis showed that the platelet count was lower in the non-survivors than in the survivors. The risk of bleeding complications may be augmented in patients with a lower level of platelet count by receiving antiplatelet therapy, affecting prognosis adversely. Conversely, the patients who underwent PCI for coronary artery disease should receive antiplatelet therapy according to the relevant guidelines to reduce the risk of stent thrombosis. Accordingly, most subjects received that therapy upon discharge, and their prognoses with and without it were not compared in this study. Therefore, the possibility that the antiplatelet therapy upon discharge had affected prognosis to some extent in this study cohort cannot be excluded. We have added this point to the limitation section (lines 279-288).
Comment 3. Please add a separate paragraph highlighting the novelty of this study, since the parameters considered for risk assessment are quite well acknowledged.
Response 3: I appreciate your helpful comment. As you mentioned, individual blood parameters considered for the risk assessment in the present study are known useful markers for predicting prognosis in patients with AMI (this phrase is newly added to the third paragraph of the discussion section: lines 224 - 225). We believe that our study's strengths and novelty were that we developed the risk-score model showing the predictive ability comparable to the GRACE 2.0 model by combining only four blood parameters, each of which has prognostic evidence in patients with AMI. Therefore, we have added a separate paragraph highlighting this point to the discussion section (lines 261- 270).
Comment 4. In table 1 please replace ‘laboratory’ with ‘laboratory parameters’ and ‘antiplatelet’ with ‘antiplatelet therapy’.
Response 4. Thank you for the suggestion. We have revised them.
Reviewer 3 Report
The manuscript is clear and well written, however there are few minor language shortcomings - critical reading by the native speaker is recommended. Also few colloquial terms like "blood sugar" should be avoided and changed for glucose.
The citations should be divided by comas, not periods.
Author Response
Response to the Reviewer 3’s Comments
Firstly, we are very grateful for the opportunity to revise our manuscript. We have carefully revised the manuscript according to your valuable comments and suggestions.
Comment 1. The manuscript is clear and well written, however there are few minor language shortcomings - critical reading by the native speaker is recommended.
Response 1: Thank you for the important suggestion. The entire manuscript has been edited and proofread by a native English speaker with expertise in the relevant medical area.
Comment 2.
Also few colloquial terms like "blood sugar" should be avoided and changed for glucose.
Response 2. Thank you for the important suggestion. We have revised "blood sugar" to "glucose."
Comment 3.
The citations should be divided by comas, not periods.
Response 3. Thank you for the helpful suggestion. We have revised the citations.
Round 2
Reviewer 2 Report
In the revised version of their manuscript the authors have adequately addressed the issues raised during the previous review. I therefore recommend the article for publication.
Author Response
Thank you very much for your kind suggestion. We have corrected some English sentences. Additionally, we attached the certificate of English proofreading in this revise submission.